# Augmenting Language Models with Long-Term Memory

**Weizhi Wang**[†], **Li Dong**[‡], **Hao Cheng**[‡], **Xiaodong Liu**[‡],
**Xifeng Yan**[†], **Jianfeng Gao**[‡], **Furu Wei**[‡]
[†]University of California, Santa Barbara   [‡]Microsoft Research
weizhiwang@ucsb.edu, {lidong1, chehao, xiaodl}@microsoft.com

## Abstract

Existing large language models (LLMs) can only afford fix-sized inputs due to the input length limit, preventing them from utilizing rich long-context information from past inputs. To address this, we propose a framework, Language Models Augmented with **Long**-Term **Mem**ory (LONGMEM), which enables LLMs to memorize long history. We design a novel decoupled network architecture with the original backbone LLM frozen as a memory encoder and an adaptive residual side-network as a memory retriever and reader. Such a decoupled memory design can easily cache and update long-term past contexts for memory retrieval without suffering from memory staleness. Enhanced with memory-augmented adaptation training, LONGMEM can thus memorize long past context and use long-term memory for language modeling. The proposed memory retrieval module can handle unlimited-length context in its memory bank to benefit various downstream tasks. Typically, LONGMEM can enlarge the long-form memory to 65k tokens and thus cache many-shot extra demonstration examples as long-form memory for in-context learning. Experiments show that our method outperforms strong long-context models on ChapterBreak, a challenging long-context modeling benchmark, and achieves remarkable improvements on memory-augmented in-context learning over LLMs. The results demonstrate that the proposed method is effective in helping language models to memorize and utilize long-form contents. Our code is open-sourced at https://aka.ms/LongMem.

## 1 Introduction

Large language models (LLMs) have revolutionized natural language processing with great successes in advancing the state-of-the-art on various understanding and generation tasks [DCLT19, RWC[+]19, LOG[+]19, YDY[+]19, BMR[+]20, RSR[+]20]. Most LLMs benefit from self-supervised training over large corpora via harvesting knowledge from fix-sized local context, showing emergent abilities, *e.g.,* zero-shot prompting [RWC[+]19], in-context learning [BMR[+]20], and Chain-of-Thought (CoT) reasoning [WWS[+]22]. Nevertheless, the input length limit of existing LLMs prevents them from generalizing to real-world scenarios where the capability of processing long-form information beyond a fix-sized session is critical, *e.g.,* long horizontal planning.

To address the length limit issue, the most straightforward method is to simply scale up the input context length. For instance, GPT-3 [BMR[+]20] increases the input length from 1k tokens in GPT-2 [RWC[+]19] to 2k tokens, thereby allowing for better capture of long-range dependencies. However, this approach typically incurs computation-intensive training from scratch and the *in-context* dense attention is still heavily constrained by the quadratic computation complexity of Transformer self-attention [VSP[+]17]. Another recent line of work [BPC20, ZGD[+]20] instead focuses on developing in-context sparse attention to avoid the quadratic cost of self-attention, which

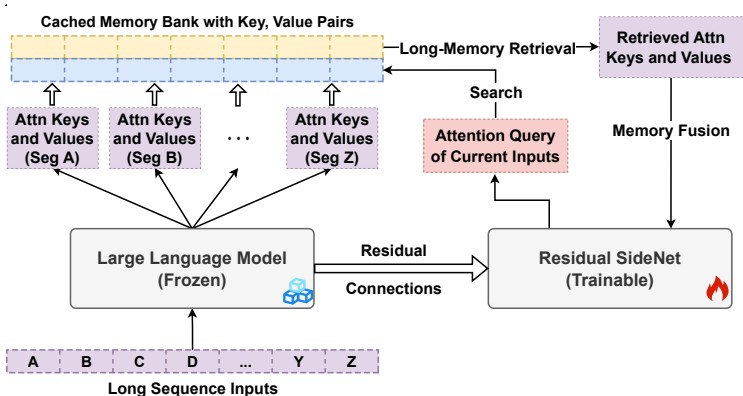

Figure 1: Overview of the memory caching and retrieval flow of LONGMEM. The long text sequence is divided into fix-length segments with each previous segment processed through a frozen backbone LLM and the corresponding attention key and value vectors of $m$-th layer are cached into the memory bank. Given current inputs, the corresponding attention query vectors are used to retrieve the top-$K$ attention key-value pairs from previous segments stored in the memory bank, which will be then fused with local context for language modeling.

still largely requires training from scratch. In contrast, the prominent work, Memorizing Transformer (MemTRM) [WRHS22], approximates in-context sparse attention via dense attention over both in-context tokens and memorized tokens retrieved from a non-differentiable memory for Transformers. Thus, MemTRM scales up the resulting language model to handle up to 65k tokens and achieves substantial perplexity gains in modeling full-length books or long papers. However, MemTRM faces the *memory staleness* challenge during training due to its coupled memory design, which uses a single model for encoding memory and fusing memory for language modeling. In other words, as the model parameters are updated, cached older representations in memory may have distributional shifts from those from the latest model, thereby limiting the effectiveness of the memory augmentation.

In this paper, we present a framework for Language Models Augmented with **Long**-Term **Mem**ory (LONGMEM). This framework enables language models to cache lengthy previous context or knowledge into a non-differentiable memory bank, and then utilize them via a decoupled memory module to mitigate the issue of memory staleness. To achieve decoupled memory, we design a novel residual side-network (SideNet) in conjunction with a frozen backbone LLM. Paired attention keys and values of the previous context are extracted using the frozen backbone LLM, which are subsequently stored in the memory bank. In the memory-augmented layer of SideNet, the generated attention query of the current input is used to retrieve cached key-value pairs of previous contexts from the memory, and the corresponding memory augmentations are then fused into adaptable hidden states via a joint-attention mechanism. Furthermore, newly designed cross-network residual connections between the SideNet and the frozen backbone LLM facilitate better knowledge transfer from the pretrained backbone LLM. Through continuous training of the residual SideNet to retrieve and fuse memory augmentations, the pre-trained LLM can be adapted to effectively leverage long-contextual memory for enhanced modeling. The detailed memory cache, retrieval and fusion process is illustrated in Figure 1.

Our decoupled memory design offers two key advantages. First, our proposed architecture effectively separates the process of encoding previous inputs into memory and the process of memory retrieval and fusion, thanks to the decoupled frozen backbone LLM and SideNet. In this way, the backbone LLM only works as the long-context knowledge encoder, while the residual SideNet works as the memory retriever and reader, which effectively resolves the issue of memory staleness. Second, directly adapting the entire LLM with memory augmentations is computationally inefficient and also prone to catastrophic forgetting. As the backbone LLM is frozen during the efficient memory-augmented adaptation stage, LONGMEM can not only tap into the pretrained knowledge but also avoid catastrophic forgetting.

LONGMEM is capable of taking various types of long-form text and knowledge into the memory bank based on downstream tasks. Here, we consider two representative cases, language modeling with full-length book contexts, and memory-augmented in-context learning with thousands of task-relevant

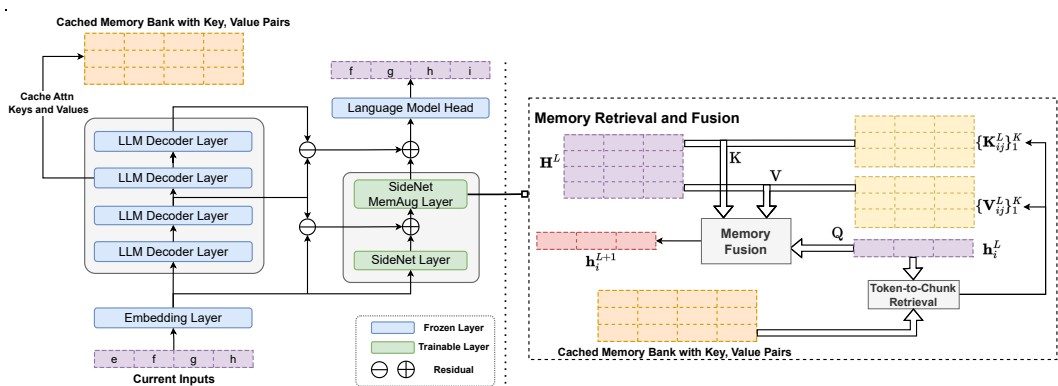

Figure 2: Overview of LONGMEM architecture. "MemAug" represents Memory-Augmented Layer.

demonstration examples. Specifically, we evaluate the effectiveness of the proposed LONGMEM on various long-text language modeling, and memory-augmented in-context learning for natural language understanding (NLU) tasks. Experimental results demonstrate that our model consistently outperforms the strong baselines in terms of long-text modeling and in-context learning abilities. Our method substantially improves LLM's long-context language modeling capabilities, with a reduction in perplexity of 1.38~1.62 over different length splits of Gutenberg-2022 corpus. Notably, our model achieves state-of-the-art performance of 40.5% identification accuracy on ChapterBreak, a challenging long-context modeling benchmark, significantly surpassing existing strong x-former baselines. Finally, with 2k demonstration examples in memory, LONGMEM shows pronounced improvements in in-context learning on popular NLU tasks, compared with prominent memory-augmented and non-memory-augmented baselines.

## 2 Methods

To enable LLMs to harvest relevant information from the past long context in memory, we propose to augment the frozen backbone LLM with a decoupled memory module. To fuse the memory context information, we design a novel lightweight residual SideNet, which can be continually trained in an efficient way. In the following, we first discuss the problem formulation of language modeling with memory augmentations. Then, we formally introduce our efficient residual SideNet for adapting the frozen pretrained LLM to jointly attend over local input context and retrieved memory context. Lastly, we provide our designed processes of how past memory is encoded, stored, recalled and fused for language modeling.

### 2.1 Language Models Augmented with Long-Term Memory

Here, we focus on the high-level problem setup and defer more component details to later sections. Given its wide adoption for pretrained LLMs, our LONGMEM model is built on the Transformer architecture [VSP+17]. For LONGMEM, there are three key components: the frozen backbone LLM, SideNet, and Cache Memory Bank. As most existing pretrained LLMs can only take a fix-sized input, only the input segment of a long sequence (*e.g.,* a book) that can fit in the length limit is denoted as the current input as done for most existing autoregressive language models. Those previous segments that can not fit are denoted as previous inputs, which are used for memory augmentations. To tap into the learned knowledge of the pretrained LLM, both previous and current inputs are encoded using the frozen backbone LLM but different representations are extracted. For previous inputs, the key-value pairs from the Transformer self-attention at $m$-th layer are stored in Cache Memory Bank, whereas the hidden states from each LLM decoder layer for the current inputs are retained and transferred to SideNet. For each current input token, top relevant key-value vector pairs are retrieved as memory augmentations for language modeling. The SideNet module can be viewed as an efficient adaption model that is trained to fuse the current input context and relevant cached previous contexts in the decoupled memory.

Formally, for a fix-sized input text sequence $\{\mathbf{x}_i\}_{i=1}^{|x|}$ (the current input), LONGMEM first performs a forward pass using the backbone LLM (indicated in blue in Figure 2) **without any gradient calculation**. The embedding layer of the backbone LLM first encodes the input $\{\mathbf{x}_i\}_{i=1}^{|x|}$ into embedding space and outputs the initial hidden states, $\mathbf{H}_{\mathrm{LLM}}^0 \in \mathbb{R}^{|x| \times E}$, where $E$ is the hidden dimension. Then each successive Transformer decoder layer of the frozen backbone LLM computes the new hidden states using the hidden states from the previous layer, $\mathbf{H}_{\mathrm{LLM}}^{l'} = f_{\theta_{\mathrm{LLM}}^{l'}}(\mathbf{H}_{\mathrm{LLM}}^{l'-1}), \forall l' \in [1, L']$ and $L'$ is the total # layers for the backbone LLM. During the forward pass with the backbone LLM for all previous inputs, the key-value pairs used for self-attention at the $m$-th Transformer decoder layer are stored in Cached Memory Bank (highlighted in orange in upper-left of Figure 2). These pairs are subsequently recalled as memory augmentations for future inputs.

**Cached Memory Bank** is a head-wise vector queue $\mathcal{Z}_k, \mathcal{Z}_v \in \mathbb{R}^{H \times M \times d}$, which maintains attention key-value pairs of latest $M$ previous inputs $\widehat{\mathbf{K}}, \widetilde{\mathbf{V}} \in \mathbb{R}^{H \times |x| \times d}$, where $H, d$ denotes the number of attention heads and per-head dimension respectively. After memory retrieval and fusion (§2.3), the memory bank removes the key-value pairs of the oldest sequences and appends the current sequences to the cached vector bank. This update mechanism ensures the language modeling causality at the sequences level and enables the memory bank to consistently maintain records of the most recent previous context for the current inputs.

After the forward pass with the backbone LLM, the SideNet module then takes all current input hidden states from the backbone LLM $\{\mathbf{H}_{\mathrm{LLM}}^{l'}\}_{l'=1}^{L'}$ and the past key-value pairs in the Cached Memory Bank for computing memory-augmented representations. Specifically, our SideNet of LONGMEM consists of $(L-1)$ normal Transformer decoder layers and one special memory-augmented decoder layer. For efficient purposes, we mainly consider the case where #layers $L$ of the SideNet is smaller than that of the backbone LLM, *i.e.*, $L < L'$. Our SideNet encodes $\mathbf{H}^0$ into memory-augmented contextual representation via $(L-1)$ normal Transformer decoder layers and a special **memory-augmented layer**.

The **memory-augmented layer** is an extension of the vanilla Transformer decoder layer that takes a memory-augmented input, including both top relevant key-value pairs in memory and the hidden states from the current input. Here, the cached key-value pairs are recalled using a token-based memory retrieval module (§2.3). For each current input token, the memory retrieval module $s_{rt}(:)$ retrieves top-$K$ relevant key-value pairs in the memory bank $\{\widetilde{\boldsymbol{k}}_{ij}, \widetilde{\boldsymbol{v}}_{ij}\}_{j=1}^K = s_{rt}(\mathbf{x}_i)$. Then SideNet computes the output using the memory-augmented input, $\mathbf{H}_{\mathrm{Side}}^{m_s} = f_{\theta_{\mathrm{Mem}}}(\mathbf{H}_{\mathrm{Side}}^{m_s-1}, \{\{\widetilde{\mathbf{k}}_{ij}, \widetilde{\mathbf{v}}_{ij}\}_{j=1}^K\}_{i=1}^{|x|})$, where $m_s$ is the layer index where we inject the memory-augmentation layer.

Finally, the token probability is computed using the last SideNet hidden states $P(\mathbf{x}_i|\mathbf{x}_1, \cdots, \mathbf{x}_{i-1}) = \mathrm{softmax}(W\mathbf{H}^L)$, where $W$ is the frozen output embedding weight shared by both the backbone LLM and SideNet. We perform a memory-augmented adaptation training for LONGMEM to utilize the decoupled memory. Following the *generative unsupervised pre-training* [RNSS18], the training objective of LONGMEM is the standard left-to-right language modeling objective, which maximizes the likelihood of the next token based on the left context: $\max \sum_{x \in \mathcal{D}} \sum_{i=1}^{|x|} \log P(\mathbf{x}_i|\mathbf{x}_1, \cdots, \mathbf{x}_{i-1})$, where $x$ is a randomly sampled sentence from the pre-training text corpus $\mathcal{D}$.

## 2.2 Residual SideNet

**SideNet Architecture and Initialization.** Here, we implement SideNet based on Transformer [VSP+17]. Specifically, the number of decoder layers $L$ in SideNet is equal to the number of layers $L'$ in the backbone LLM divided by a reduction factor (a layer reduction factor of 2 is used throughout this work, *i.e.*, $L' = 2L$). The weights of each decoder layer in SideNet are initialized from the corresponding pre-trained decoder layer of the backbone LLM at the same depth: $\Theta_{\mathrm{Side}}^l = \Theta_{\mathrm{LLM}}^{2l}$. As illustrated in Figure 2, the SideNet model takes the output of backbone LLM's embedding layer and reuses the language modeling head of the backbone LLM, which remains frozen during the continual adaption stage. Throughout the memory-augmented adaptation stage, all other parameters of SideNet are updated based on the training signal. In this way, the lightweight SideNet adaptation achieves fast convergence with knowledge transferred from pre-trained parameters.

**Cross-Network Residual Connections.** To tap into knowledge from the pretrained backbone LLM, we utilize our proposed cross-network residual connections to fuse representations from the backbone

LLM into SideNet. Specifically, we add the difference between output hidden states at $2l$-th and $(2l-2)$-th layers of the backbone LLM as the residual connections to the output hidden states at $l$-th layer of SideNet. Then, the input to the next $(l+1)$-th layer of SideNet is the sum of the original hidden state forwarded through the previous layer $f_{\Theta_{\text{Side}}^l}(\mathbf{H}_{\text{Side}}^{l-1})$ and the cross-network residual connection of the hidden state difference from the backbone LLM

$$\mathbf{H}_{\text{Side}}^l = f_{\Theta_{\text{Side}}^l}(\mathbf{H}_{\text{Side}}^{l-1}) + (\mathbf{H}_{\text{LLM}}^{2l} - \mathbf{H}_{\text{LLM}}^{2l-2}), \forall l \in [1, L], \tag{1}$$

where $\mathbf{H}^0$ is the output of embedding layer. It is worth noting that the residual connections after the self-attention and feed-forward network of a decoder layer [VSP$^+$17] will be performed as normal in $f_{\Theta_{\text{Side}}^l}(\mathbf{H}_{\text{Side}}^{l-1})$ and parallel to the proposed cross-network residual connections.

## 2.3  Memory Retrieval and Fusion

The long-term memory capability of LONGMEM is achieved via a memory-augmentation module for retrieval and fusion.

**Token-to-Chunk Memory Retrieval.** Instead of performing token-to-token retrieval, we focus on token-to-chunk retrieval for acceleration and integrity. A text-chunk refers to an n-gram structure of chunk-size $csz$ number of contiguous tokens. The memory bank stores cached key-value pairs at the level of token chunks. We divide the memory bank into $M/csz$ attention key-value paired chunks and use the mean-pooled vector on the chunk-size dimension to get the key vector for retrieval. Then we retrieve the top-$(K/csz)$ attention key-value chunks w.r.t the dot product between the attention query of the current input token and the mean-pooled attention key of a candidate chunk. Finally, we squeeze the chunk-size dimension for retrieved key-value paired chunks and flatten them into $K$ key-value pairs at token-level $\{\widehat{\mathbf{K}}_j, \widetilde{\mathbf{V}}_j\}_{j=1}^K$. Adopting token-to-chunk retrieval reduces the size of the retrieval index and accelerates the process. Meanwhile, the retrieval accuracy can be further improved, which is also observed in [LGW$^+$23] and [BMH$^+$21]. The hyperparameter chunk-size $csz$ controls the granularity of retrieved contexts, which can be empirically adjusted based on downstream tasks. For instance, in-context learning requires more fine-grained label tokens from demonstration examples cached in memory, where a smaller $csz$ is helpful.

**Memory Fusion.** The memory fusion is performed within a special memory-augmented layer. As the conventional Transformer decoder layer uses the multi-head self-attention [VSP$^+$17], we follow [WRHS22] to extend it to a joint-attention mechanism and propose a long-term memory fusion process to enable each token to attend on both local contexts and retrieved memory contexts. With the head-wise hidden state output from previous layer $\mathbf{H}^{l-1} \in \mathbb{R}^{|x| \times d}$ and the corresponding retrieved attention key-value pairs are $\{\widetilde{\mathbf{K}}_i, \widetilde{\mathbf{V}}_i\}_{i=1}^{|x|} \in \mathbb{R}^{|x| \times K \times d}$, the output hidden state for the $l$-th memory-augmented layer $\mathbf{H}^l$ is computed as:

$$\mathbf{A} = \text{softmax}(\frac{\mathbf{Q}\mathbf{K}^T}{\sqrt{d}})\mathbf{V}, \quad \mathbf{M} = \text{Concat}\{\text{softmax}(\frac{\mathbf{Q}_i\widetilde{\mathbf{K}}_i^T}{\sqrt{d}})\widetilde{\mathbf{V}}_i\}_{i=1}^{|x|}, \tag{2}$$

$$\mathbf{H}^l = \text{sigmoid}(g) \cdot \mathbf{A} + (1 - \text{sigmoid}(g)) \cdot \mathbf{M}, \tag{3}$$

where $\mathbf{Q}, \mathbf{K}, \mathbf{V}, \mathbf{A}, \mathbf{M} \in \mathbb{R}^{|x| \times d}$, K is the number of retrieved attention key-value pairs in cached memory for each token, and $g$ is a trainable head-wise gating vector. The hidden state output from previous layer $\mathbf{H}^{(l-1)}$ is linearly projected into attention queries, keys, and values $\mathbf{Q}, \mathbf{K}, \mathbf{V}$ separately via three matrices $W^Q, W^K, W^V \in \mathbb{R}^{d \times d}$. It is worth noting that the retrieved attention key-value pairs in cached memory are distinct to each token.

## 3  Experiments

We evaluate our proposed LONGMEM model on different tasks that require long-context modeling: a) long-text language modeling and language understanding when loading the past long-context into cached memory; b) infinite-length in-context learning when loading a large number of demonstration examples into cached memory.

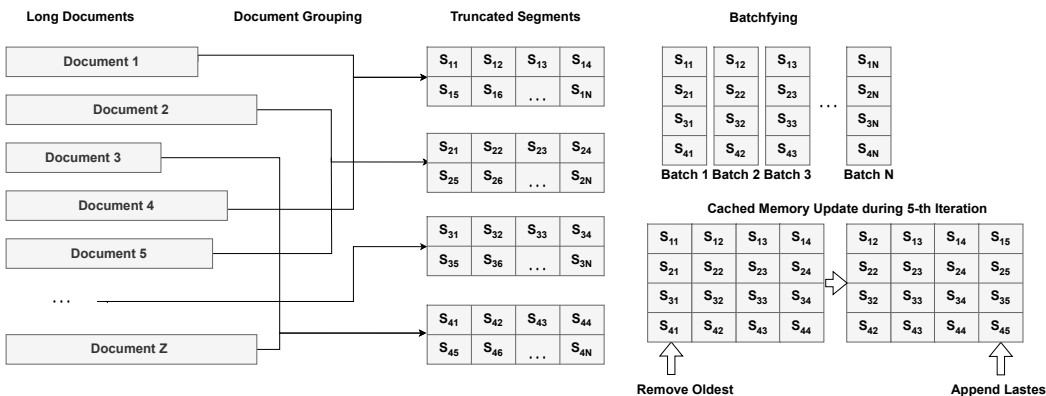

Figure 3: Batchfying the large text corpora into batches to ensure that each consecutive segments within each document is distributed in consecutive batches.

## 3.1 Training Setup

**Batchfying the training corpora.** The conventional batchyfing process for large corpora truncates the whole corpora into consecutive fix-length text segments without padding and shuffles all segments to construct mini-batches [RWC+19]. In contrast, LONGMEM must disable global shuffling and ensure the global causality at the segment level. Firstly, we divide all long documents in training corpora into batch-size number of document groups with equivalent length and then perform a document-level shuffling within each group. Then, we concatenate shuffled documents within one group and truncate them into ordered segments. In order to ensure that two consecutive segments of one long document are distributed in two consecutive input batches after batchfying, we select one segment from batch-size number of document groups with the same inner-group index. Thus a mini-batch with batch-size number of segments are constructed from exactly the batch-size number of document groups. In this way, as the training iteration steps, the cached attention key-value pairs in the memory bank are previous context of current inputs within the same document. The batchfying process is illustrated in Figure 3.

**Training Corpus, Backbone LLM and Hyperparameter.** We sample a subset of the Pile [GBB+20] as the training corpus, including BookCorpus2, Books3, OpenWebText2, Stack Exchange, Wikipedia, Gutenberg (PG-19), NIH ExPorter, and Pile-CC datasets. We reproduce GPT-2* (407M-params) as the pre-trained backbone LLM with Alibi [PSL21] position embedding because original GPT-2 [RWC+19] adopts absolute position embedding, which is found to perform poorly to enable LLM to learn long-distance dependencies [DYY+19]. The backbone LLM holds a $L' = 24, H = 16, d = 64$ architecture. The SideNet holds a $L = 12, H = 16, d = 64$ architecture. The training for memory-augmented adaptation iterates on 26B tokens, with a global 256 batch-size and 1024 sequence length. The chunk-size $csz$ is 4 tokens and the memory size $M$ is 65k key-value pairs of tokens. For each token, we retrieve $K$=64 attention key-value pairs for augmentation, which are $K/csz$=16 text chunks. The memory-augmentation layer is the 9-th layer of SideNet. The attention keys and values from 18-th layer of backbone LLM is cached into memory and used for future retrieval. Other training details are presented in Appendix C.

**Memory Retrieval Module.** The fixed memory-size of cached memory bank in one GPU is 65536 key-value pairs of tokens. We enable each GPU to construct and update their own memory retrieval module for efficiency. For the implementation of the efficient token-to-chunk retrieval, we use the `faiss` [JDJ21] toolkit to construct an exact-search index on GPU to store the mean-pooled attention keys of text chunks and perform efficient retrieval. The `faiss` index maintains a fixed $M/csz$ keys and provides the efficient exact search w.r.t. inner product. The retrieval takes about 15ms per 1k tokens, which is 55% timecost of backbone LLM forwarding pass. We can easily adapt the exact search index to approximate search index to gain more retrieval efficiency.

**Baselines.** In addition to the baseline of our pre-trained GPT-2* variant, we consider Memorizing Transformer (MemTRM) [WRHS22] and TRIME [ZLC22] as two memory-augmented baselines. The MemTRM model can be easily adapted to tune a pre-trained LLM to use external memory. We insert the KNN-augmented layer proposed by MemTRM as the same 18-th layer in the LLM decoder.

| Dataset | PG-22 | | | | | ArXiv |
| Splits | S1 | S2 | S3 | S4 | S5 | |
| --- | --- | --- | --- | --- | --- | --- |
| **Len. Range** | 5K-10K | 10K-100K | 100K-500K | 500K-1M | >1M | <60K |
| **#Documents** | 500 | 100 | 30 | 8 | 1 | 100 |
| **Avg. #tokens** | 7.6K | 47.6K | 140K | 640K | 1.2M | 15.4K |

Table 1: Dataset Statistics of five splits of PG-22 based on length range and ArXiv.

To adapt TRIME for our experiments, we replace the batchfying function and loss function of training GPT-2* with those proposed by TRIME, which enables a memory-augmented adaptation tuning method for LLMs. The two reproduced baselines are trained for the same number of tokens under the same hyperparameter setting as LONGMEM.

## 3.2 Long-Context Language Modeling

The long-context language modeling can potentially benefit from the augmented memory of past long-contexts. The knowledge stored in retrieved attention key-values can provide valuable background and contextual information, helping models perform better in long-context language modeling. For instance, when trying to model a long-text book, acquiring knowledge from previous background and character relationships can be helpful in modeling the subsequent stories.

**Evaluation Setting.** We first compare LONGMEM and baselines on three long-context modeling datasets, *Project Gutenberg 2020-2022*, *ArXiv*, and *ChapterBreak*. The majority of included books or papers in these datasets have the length of at least 16k tokens. All listed datasets are evaluated in a **zero-shot** manner without any task-specific tuning. The detailed evaluation settings on the three datasets are as follows:

- **Project Gutenberg 2020-2022 Language Modeling Dataset.** We crawled and cleaned the books published between 2020 and 2022 under Project Gutenberg Library[1] to build up a completely new long-text modeling dataset, named **PG-22**. It is significantly different from our training subset PG-19 in terms of domains and writing styles, because books in PG-19 [RPJL19] are published before 1919. We provide different validation splits of PG-22 based on length range, and the data statistics are presented in Table 1.

- **ArXiv Dataset.** The ArXiv dataset includes papers in the areas of Math, Computer Science, and Physics. We select a validation split of ArXiv paper subset in the Pile corpus [GBB+20]. The ArXiv subset of Pile is excluded from our training and serves an out-of-distribution dataset. We report the token-level language modeling perplexity on the long-context language modeling benchmarks of PG-22 and ArXiv.

- **ChapterBreak Benchmark.** ChapterBreak [STI22] is a challenging suffix identification dataset that requires LLMs to distinguish the beginning of the ground-truth next chapter from a set of hard negative segments sampled from the same book, given the long context of previous chapters. ChapterBreak requires processing global long-context to comprehend and identify the correct suffix. [STI22] demonstrated that even state-of-the-art x-formers for long-text processing fail to effectively leverage long-range context to perform well on ChapterBreak. ChapterBreak has two subsets, the PG-19 subset and the Archive of Our Own (AO3) subset. As the PG-19 corpus has been included in the pre-training corpus of LONGMEM, it cannot be further used for evaluation. Thus, we select AO3 subset, which contains fan-fictions extracted from AO3. ChapterBreak provides 8 splits based on the prefix length from 0.5k to 8k tokens to fit the length limit of different models. The splits of 4k, 6k, and 8k prefix are selected for evaluation. For LLMs that cannot process over 4k tokens, we abandon the front prefix to fulfill the maximum input length of LLMs. For memory-augmented models (MemTRM and LONGMEM), we load the given 4k/6k/8k prefix contexts into the cached memory and then do the scoring. we use the perplexity as the scorer for each candidate suffix segment in a zero-shot manner. Then the suffix segment with lower perplexity is selected as the label. The suffix identification accuracy is used as the evaluation metric.

**Results.** The main results on evaluated long-context datasets are summarized in Table 2. The proposed LONGMEM model significantly outperforms all considered baselines on long-text language modeling

---

[1]https://www.gutenberg.org/

| Model | In-Context Len. | In-Memory Len. | PG-22 | | | | | ArXiv |
|---|---|---|---|---|---|---|---|---|
| | | | 5K-10K | 10K-100K | 100K-500K | 500K-1M | >1M | |
| GPT-2* | 1k | N/A | 22.78 | 24.39 | 24.12 | 24.97 | 18.07 | 11.05 |
| MemTRM | 1k | 65K | 21.77 | 23.56 | 23.23 | 24.16 | 17.39 | 10.81 |
| TRIME | 1k | 65K | 22.21 | 23.50 | 23.74 | 24.32 | 17.80 | 10.95 |
| LONGMEM | 1k | 65K | **21.29** | **23.01** | **22.55** | **23.35** | **16.71** | **10.05** |

Table 2: Evaluation results on long-context language modeling datasets. We report token-level perplexity (PPL) (lower the better) on all datasets.

| Model | #Params | In-Context Len. | In-Memory Len. | ChapterBreak$_{ao3}$ | | |
|---|---|---|---|---|---|---|
| | | | | ctx-4k | ctx-6k | ctx-8k |
| GPT-2-XL[†] [RWC[+]19] | 1.5B | 1K | N/A | 24% | 24% | 24% |
| GPT-3[†] [BMR[+]20] | 175B | 2K | N/A | 28% | 28% | 28% |
| LocalTRM[†] [RSVG21] | 516M | 8K | N/A | 24% | 24% | 24% |
| RoutTRM[†] [RSVG21] | 490M | 8K | N/A | 25% | 24% | 24% |
| Bigbird[†] [ZGD[+]20] | 128M | 4K | N/A | 26% | 26% | 26% |
| GPT-2* | 407M | 1K | N/A | 18.4% | 18.4% | 18.4% |
| MemTRM | 407M | 1K | $\infty$ | 28.3% | 28.7% | 28.7% |
| LONGMEM | 558M | 1K | $\infty$ | **37.7%** | **39.4%** | **40.5%** |

Table 3: Zero-shot Suffix Identification Accuracy on AO3 subset of `ChapterBreak`. Baselines marked with [†] are directly cited from [STI22]. The MemTRM and LONGMEM loads the given 4k/6k/8k prefix contexts into cached memory, while the input length to local context is still 1k tokens.

datasets, with improvements of 1.38 to 1.62 perplexity on different length splits of *PG-22*, and 1.0 on ARXIV datasets. Surprisingly, the proposed method achieves the state-of-the-art performance of 40.5% accuracy on `ChapterBreak`$_{AO3}$ suffix identification benchmark and outperforms both the strong long-context transformers and GPT-3 with 313x larger parameters. The substantial improvements on these datasets demonstrate that LONGMEM can comprehend past long-context in cached memory well for predicting future inputs.

### 3.3 Memory-Augmented In-Context Learning

LLMs have the emerging capability of in-context learning (ICL) via learning knowledge non-parametrically from few-shot demonstration examples in the local context. However, conventional in-context learning is heavily restricted by input context length, rendering it ineffective to absorb supervision from sufficient demonstration examples in the training set. With the proposed unlimited-length memory augmentation, LONGMEM can overcome the limitation of the number of demonstration examples in the local context and even attend on the whole training set by loading it into the cached memory. In this way, LONGMEM generalizes the conventional few-shot in-context learning to memory-augmented in-context learning with thousands of auxiliary demonstration examples.

**Evaluation Setting.** Here, we evaluate the in-context learning capability of baselines and the proposed LONGMEM model on five NLU datasets, SST-2 [SPW[+]13], MPQA [WWC05], MR [ABK[+]07], Subj [PL04] and SST-5 [SPW[+]13]. We evaluate models on two few-shot settings, 4-shot and 20-shot. The 4-shot case is the data-insufficient scenario, while the 20-shot demonstrations can almost fulfill the 1k input length and provide sufficient contextual self-supervisions. We transform the k-shot examples to semantically meaningful demonstration examples via fixed text template, i.e., $d_i$="Review: $x_i$ Sentiment: $y_i$",$\forall\{(x_i, y_i)\}_{i=1}^{k} \in \mathcal{D}_{\text{train}}$ for sentiment analysis tasks. Additionally, we evaluate the 3-shot ICL on question-answering using SQuAD [RZLL16] under an open-ended generation setting. The details of all prompt templates are presented in Appendix D. Then we concatenate the demonstration examples with newlines to delimit them. The prediction label is directly generated using greedy decoding given the demonstration examples and test cases in context. The prediction accuracy is used as the evaluation metric. We report the mean and standard deviation of 6 runs with different random seeds to assess the randomness in selecting k-shot demonstration examples. As mentioned previously, the chunk size controls the granularity of retrieved text chunks. Since the considered NLU datasets require more fine-grained labels from cached memory, we perform

| Model | In-Context #Demons. | In-Memory #Demons. | SST-2 ACC↑ | MR ACC↑ | Subj ACC↑ | SST-5 ACC↑ | MPQA ACC↑ | Avg. |
|---|---|---|---|---|---|---|---|---|
| Majority | N/A | N/A | 50.9 | 50.0 | 50.0 | 20.0 | 50.0 | 44.2 |
| GPT-2* | 4 | N/A | $68.3_{11.6}$ | $64.7_{12.5}$ | $51.9_{4.2}$ | $31.4_{4.4}$ | $61.5_{11.8}$ | 55.6 |
| MemTRM | 4 | 2000 | $67.5_{12.4}$ | $64.6_{11.3}$ | $53.2_{6.0}$ | $29.6_{4.4}$ | $63.0_{12.1}$ | 55.6 |
| TRIME | 4 | 2000 | $69.5_{14.5}$ | $63.8_{9.8}$ | $51.5_{1.5}$ | $31.8_{6.7}$ | $63.6_{12.9}$ | 56.0 |
| LONGMEM | 4 | 2000 | $\mathbf{71.8}_{14.0}$ | $\mathbf{65.1}_{11.0}$ | $\mathbf{53.8}_{3.7}$ | $\mathbf{36.0}_{6.8}$ | $\mathbf{65.4}_{12.8}$ | **58.4** |
| GPT-2* | 20 | N/A | $68.2_{11.5}$ | $63.4_{5.2}$ | $57.6_{10.2}$ | $33.6_{6.0}$ | $70.8_{7.6}$ | 58.7 |
| MemTRM | 20 | 2000 | $65.1_{9.6}$ | $65.1_{9.3}$ | $58.2_{10.6}$ | $31.9_{6.3}$ | $72.7_{7.4}$ | 58.6 |
| TRIME | 20 | 2000 | $74.3_{13.9}$ | $71.5_{2.5}$ | $57.5_{11.4}$ | $33.0_{4.6}$ | $69.8_{7.8}$ | 61.1 |
| LONGMEM | 20 | 2000 | $\mathbf{78.0}_{14.1}$ | $\mathbf{78.6}_{3.3}$ | $\mathbf{65.6}_{8.5}$ | $\mathbf{36.5}_{7.5}$ | $\mathbf{74.6}_{7.3}$ | **66.7** |

Table 5: Accuracy [%] of 4-shot and 20-shot ICL on 5 NLU tasks (SST-2, mr, subj, SST-5, mpqa). We sample 2000 extra demonstration examples and load them into cached memory. The subscript is the standard deviation across 6 runs. Avg. refers to the average accuracy on 5 datasets.

a hyperparameter selection on the validation set of SST-2, and the best chunk-size 2 is used to report the results for MemTRM, TRIME and our model.

**Results.** The results on in-context learning are summarized in Table 5 and Table 4. LONGMEM achieves remarkable improvements on all NLU tasks under the 20-shot sufficient in-context setting, with +5.6 average scores increase over pretrained GPT-2*, MemTRM, and TRIME. Meanwhile, LONGMEM also brings performance improvements on the 4-shot case. Additionally, LONGMEM improves the in-context learning capabilities of LLMs on open-ended generation tasks, with +4.5 EM score increase on SQuAD. The results indicate that having more demonstration examples loaded in cached memory can provide additional contextual cues to assist in-context learning. LONGMEM can utilize task-relevant knowledge from both local contextual demonstrations and in-memory augmented demonstrations, thereby achieving superior in-context learning capabilities.

| Model | EM | F1 |
|---|---|---|
| GPT-2* | $22.28_{2.3}$ | $30.78_{2.0}$ |
| MemTRM | $22.84_{3.5}$ | $32.65_{2.8}$ |
| LONGMEM | $26.77_{2.3}$ | $35.70_{2.0}$ |

Table 4: Exact match (EM) and F1 scores of 3-shot (about 1k tokens) in-context learning on SQuAD. LONGMEM loads 200 extra demonstration examples into cached memory.

## 3.4 Ablation Studies

So far, we empirically verify the effectiveness and superiority of LONGMEM in utilizing cached memory for long-context modeling, long-context understanding, and many-shot in-context learning. Furthermore, we would like to investigate the extend to which the cached memory contributes to the long-context understanding capability of LONGMEM through an ablation study of removing memory augmentations. Besides, since the design of the cached memory bank involves several hyperparameters, such as memory size $msz$ and chunk-size $csz$, we conduct a series of ablation studies to evaluate the effects of those choices.

**Effects of Long-Term Memory Augmentation.** To evaluate the effects and contributions of memory augmentations, we set the memory-size to 0 and maintain the SideNet parameters during inference. The results of LONGMEM without memory augmentation are shown in Table 6 of Appendix. As expected, without augmented long-term memory, the vanilla model with only backbone LLM and SideNet only gains 59.4 average scores on ICL NLU tasks, which is a 7.3 average accuracy decrease due to the removal of memory augmentation.

**Effects of Chunk-Size.** As analyzed before, the chunk-size $csz$ controls the granularity of retrieval and thus it may make a difference to tasks with requirements of fine-grained retrieval. We perform an ablation study on the effects of various chunk-size choices $csz \in \{2, 4, 8\}$ for in-context learning and the results are presented in 4(a). The chunk size of 2 yields the best performance on in-context learning tasks on five NLU datasets, which is consistent with the property of NLU tasks with the requirement of fine-grained retrieval and fusion towards classification label tokens.

**Effects of Memory Size.** The memory size (msz) controls the capacity of the memory bank. In general, the memory size should be compatible with the average length of documents or contexts,

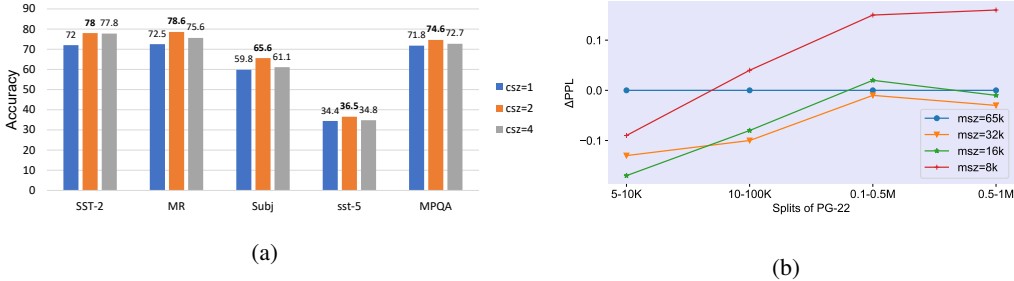

(a)                                                    (b)

Figure 4: (a) Accuracy on 5 NLU datasets given different chunk size during inference; (b) $\Delta$Perplexity on 4 splits of PG-22 given different memory size during inference, in which the perplexity when $msz$=65k is used as baseline.

*i.e.,* a set of books with average 16k tokens should deploy the memory size of 16k tokens in cached memory. The training $msz$ of 65 tokens is excessive for downstream tasks such as ChapterBreak as the whole prefix context length does not exceed 65k tokens. Thus, we perform an ablation study on the effects of memory size $msz \in \{8k, 16k, 32k, 65k\}$ during the inference stage on the PG-22 language modeling datasets and the results are shown in 4(b). To model the books with lengths of 8k-50k, the smaller memory size $16k$ which is consistent with the average length of target books yields the best perplexity.

## 4   Related Work

**Large Language Models.** Large Language Models, *i.e.,* GPT-3 [BMR$^+$20], LLAMA [TMS$^+$23], GPT-4 [Ope23], significantly revolutionized NLP research and promoted the state-of-the-art of various language understanding, language generation [WZG$^+$22], and even vision-language tasks [WDC$^+$22]. Additionally, enabled by multi-task instruction tuning [WBZ$^+$21, OWJ$^+$22], LLMs exhibit "emergent abilities" [WTB$^+$22] like mathematical reasoning [WWS$^+$22], code completion [CTJ$^+$21], etc.

**x-formers.** To enable transformers to attend on longer context, many variants of "x-formers" are proposed. Transformer-XL [DYY$^+$19] proposes to cache attention keys and values of past segment and reuse them in recurrent manner. Recent seminal works of x-formers, including LinFormer [WLK$^+$20], LongFormer [BPC20], Routing Transformer [RSVG21], proposed various sparse attention mechanisms for decreasing $O(n^2)$ complexity to $O(n \log n)$ or even $O(n)$. BigBird [ZGD$^+$20] achieves a 4k sequence length via attending on a subset of context tokens. Although these x-formers achieve substantial efficiency improvements, such efficiency gains are not remarkable when modeling sequences that spans book-level length. Moreover, the largest sequence length of these methods is still upper-bounded by 16k tokens, making them invalid in modeling long-sequences at the book or wikipedia-page level (*i.e.,* average 70k tokens for full-length books in PG19 dataset [RPJL19]).

**Side-Tuning.** The method of Side-Tuning [ZSZ$^+$20, SCB22] is a task-specific tuning method for pre-trained models via training a lightweight side-network that is fused with the fixed pre-trained network via summation. Our method inherits the idea of adopting a side-network but distinguishes the side-tuning method in terms of learning objective and cross-network fusion ways. LONGMEM proposes to augment LLMs with decoupled memory to retrain information from long past inputs without any task-specific tuning. The cross-network residual connections introduced here are novel and distinct from the vanilla summation used in Side-Tuning.

## 5   Conclusion

In this paper, we propose to augment LLMs with long-term memory for enabling them to memorize long-form context and gain long-form memory. The designed decoupled memory module can cache attention key and value pairs of past inputs for future retrieval and fusion. A decoupled residual SideNet is introduced as the memory retriever and reader, meanwhile the LLM itself is frozen and works as knowledge and memory encoder. Experiments on various long-contextual language modeling datasets demonstrate the effectiveness of our model over other memory-augmentation baselines. The proposed method can also enable in-context learning of LLMs to overcome the limited number of demonstration examples in context, which is constrained by the contextual length, via caching thousands of auxiliary demonstration examples in memory.

## Acknowledgement

This work is done during the first author's internship at Microsoft Research. We would like to thank the anonymous reviewers for the helpful comments. We appreciate Yutao Sun and Yaru Hao for helpful suggestions on implementation and evaluation benchmarks. The first author was partly sponsored by the DARPA PTG program (HR001122C0009). Any opinions, findings, conclusions, or recommendations expressed in this paper are those of the authors and do not necessarily reflect the views of funding agencies.

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

# A    Ablation Study on the Effect of Memory Augmentation

| Model | In-Context #Demons. | In-Memory #Demons. | SST-2 ACC↑ | MR ACC↑ | Subj ACC↑ | SST-5 ACC↑ | MPQA ACC↑ | Avg. |
|---|---|---|---|---|---|---|---|---|
| Majority | N/A | N/A | 50.9 | 50.0 | 50.0 | 20.0 | 50.0 | 44.2 |
| GPT-2* | 4 | N/A | $68.3_{11.6}$ | $64.7_{12.5}$ | $51.9_{4.2}$ | $31.4_{4.4}$ | $61.5_{11.8}$ | 55.6 |
| MemTRM | 4 | 2000 | $67.5_{12.4}$ | $64.6_{11.3}$ | $53.2_{6.0}$ | $29.6_{4.4}$ | $63.0_{12.1}$ | 55.6 |
| TRIME | 4 | 2000 | $69.5_{14.5}$ | $63.8_{9.8}$ | $51.5_{1.5}$ | $31.8_{6.7}$ | $63.6_{12.9}$ | 56.0 |
| LONGMEM | 4 | 2000 | $\mathbf{71.8}_{14.0}$ | $\mathbf{65.1}_{11.0}$ | $\mathbf{53.8}_{3.7}$ | $\mathbf{36.0}_{6.8}$ | $\mathbf{65.4}_{12.8}$ | **58.4** |
| w/o Memory | 4 | 0 | $69.4_{12.4}$ | $64.3_{12.1}$ | $53.4_{7.7}$ | $29.0_{5.2}$ | $62.5_{12.3}$ | 55.7 |
| GPT-2* | 20 | N/A | $68.2_{11.5}$ | $63.4_{5.2}$ | $57.6_{10.2}$ | $33.6_{6.0}$ | $70.8_{7.6}$ | 58.7 |
| MemTRM | 20 | 2000 | $65.1_{9.6}$ | $65.1_{9.3}$ | $58.2_{10.6}$ | $31.9_{6.3}$ | $72.7_{7.4}$ | 58.6 |
| TRIME | 20 | 2000 | $74.3_{13.9}$ | $71.5_{2.5}$ | $57.5_{11.4}$ | $33.0_{4.6}$ | $69.8_{7.8}$ | 61.1 |
| LONGMEM | 20 | 2000 | $\mathbf{78.0}_{14.1}$ | $\mathbf{78.6}_{3.3}$ | $\mathbf{65.6}_{8.5}$ | $\mathbf{36.5}_{7.5}$ | $\mathbf{74.6}_{7.3}$ | **66.7** |
| w/o Memory | 20 | 0 | $70.0_{12.8}$ | $70.8_{6.2}$ | $52.9_{4.6}$ | $30.9_{6.4}$ | $72.5_{7.5}$ | 59.4 |

Table 6: Ablation study results on the effect of memory augmentation of 4-shot and 20-shot ICL on 5 NLU tasks (SST-2, mr, subj, SST-5, mpqa). We sample 2000 extra demonstration examples and load them into cached memory. The subscript is the standard deviation across 6 runs. Avg. refers to the average accuracy on 5 datasets. "w/o" is short for "without".

# B    Inference Efficiency and GPU-Memory Efficiency

When the model is required to comprehend long sequences, the proposed method LONGMEM can load the out-of-boundary inputs into the cached memory as previous context. Thus, the memory usage and inference speed can be significantly improved compared with vanilla self-attention-based models. The detailed statistics in terms of the efficiency is presented in Table 7.

| Model | In-Context Len. | In-Memory Len. | Inference Speed (tokens/s)↑ | GPU-Memory Usage (MBs)↓ |
|---|---|---|---|---|
| GPT-2* | 4k | N/A | 14666 | 20671 |
| LONGMEM | 1k | 3k | 22638 | 13335 |
| GPT-2* | 8k | N/A | 8417 | 54195 |
| LONGMEM | 1k | 7k | 21343 | 13437 |

Table 7: The superiority of our method over fully dense self-attention (GPT-2*) in terms of inference speed and GPU-memory utilization.

## C  Training Details

The pre-training of reproduced GPT-2* iterates on 117B tokens in total, with 512 batch-size and 1024-token fixed segment-length. The Adam optimizer [KB15] is adopted in memory-augmented adaptation training. The pre-training and adaptation are trained on 16 32GB-Tesla-V100 GPUs. Other detailed training hypperparamters and settings are presented in Table 8.

| Hyperparameter | LONGMEM |
|---|---|
| **Reproduced GPT-2* Backbone LLM Hyperparameters** | |
| Parameters | 407M |
| Precision | `float16` |
| Layers | 24 |
| Hidden dim. | 1024 |
| Attention heads | 16 |
| Head Dim | 64 |
| Vocab size | 52k |
| Sequence length | 1024 |
| Position emb. | `Alibi` |
| Tied embedding | `False` |
| **SideNet Hyperparameters** | |
| Parameters | 151M |
| Precision | `float16` |
| Layers | 12 |
| Hidden dim. | 1024 |
| Attention heads | 16 |
| Head Dim | 64 |
| Sequence length | 1024 |
| **Memory-Augmented Adaptation Hyperparameters** | |
| Global Batch Size | 256 |
| Learning rate | 2.0e-4 |
| Total tokens | 26B |
| Warmup tokens | 0 |
| LR Decay style | polynomial |
| Adam $(\beta_1, \beta_2)$ | (0.9, 0.98) |
| Adam eps | 1e-06 |
| Weight decay | 0.01 |
| Gradient clipping | 2.0 |

Table 8: Memory-Augmented Adaptation and Architectural Hyperparameters.

## D  Prompting Templates

We present all hand-crafted in-context learning prompting templates and labels for 5 NLU datasets and Squad QA dataset in Tabel 9.

| Task | Prompt | Labels |
|---|---|---|
| **SST-2** | Review: [Sentence] Sentiment: [Label] | {positive, negative} |
| **MR** | Review: [Sentence] Sentiment: [Label] | {positive, negative} |
| **MPQA** | Review: [Sentence] Sentiment: [Label] | {positive, negative} |
| **SST-5** | input: [Sentence] type: [Label] | {terrible,bad,okay,good,great} |
| **Subj** | input: [Sentence] type: [Label] | {objective, subjective} |
| **Squad** | Passage: [Passage]\n Question: [Question] Answer: [Answer] | |

Table 9: The hand-crafted prompts used to query the model predictions on the zero-shot evaluation of 5 NLU datasets and one question-answering dataset Squad.

