# OpenReview forum: "Augmenting Language Models with Long-Term Memory"
_NeurIPS.cc/2023/Conference — NeurIPS 2023 poster_

### Official Review · Reviewer_UtJk · 2023-07-02

**Soundness:** 3 good
**Presentation:** 2 fair
**Contribution:** 3 good
**Rating:** 7
**Confidence:** 4

**Summary:**

This paper proposes an architectural extension to a frozen LLM that allows for incorporation of external memory caches (also constructed with the frozen LLM), allow the model to handle extremely long contexts.

**Strengths:**

The approach presented is sensible (although I am not sufficiently familiar with the latest long-context modeling literature to determine if it is novel), and the experiments are convincing. In particular, I think the few-shot learning results are very strong.

**Weaknesses:**

- The explanation of the model architecture is fairly unclear to me, and I found it difficult to match the description of the architecture and the presented diagram. I recommend the authors rewrite the explanation of the model and obtaining feedback from a fresh reader for suggestions on improving the clarity. Alternatively, presenting the architecture in terms of pseudo-code may also be clearer.
- The evaluations presented are largely newly proposed (besides ChapterBreak). They appear to be sensibly constructed, but it does mean the results are slightly difficult to compare to or contextualize to other works. I would be interested in seeing some experiments on more established tasks (e.g. long document summarization).

**Questions:**

- I believe GPT-2* refers to the reproduced GPT-2, but I don't think this is mentioned in the text.
- Perplexity should be a positive number. Are the authors referring to negative log loss instead?
- I would like to see experiments on with inference DEMA but with no memory cache (if I understand the model architecture correctly, this would still involve the SideNet parameters, which would differ from GPT-2*). This should apply to all current experiments.

---

> ### Author Rebuttal · Authors · 2023-08-10
>
> We thank you a lot for your detailed feedback and suggestions!
>
>
> Q1: Issues of presentation
>
> Thank you for the kind suggestion on optimizing the presentation. We follow your advice and collect some reader feedback. Then we decompose the main figure into two figures for better presentation. The first figure describes the memory encoding, caching, and retrieval flow of the proposed method. And the second figure illustrates more on the architectural contributions. You can refer to these revised figures in the one-page rebuttal PDF document attached.
>
> Q2: Evaluation on summarization tasks
>
> Thank you for the great suggestion on verifying the proposed method on language generation tasks like summarization. As the proposed DEMA model is a pre-trained language model augmented with long-term memory, the designed evaluations focus more on the emerging capabilities enabled by proposed memory-augmented pre-training, especially in-context learning or zero-shot learning. Therefore, we evaluate the proposed method on both natural language understanding and natural language generation tasks on memory-augmented in-context learning settings. The performance improvement on Squad in Table 4 can demonstrate the effectiveness of the method on language generation tasks. The document summarization or reading comprehension tasks usually require full-parameter model fine-tuning on the knowledge domain of the dataset and thus not a good fit for evaluating the capability of pre-trained language models. But we will later tune the proposed method on BookSum dataset for evaluating such domain-specific adaptation capability.
>
> Q3: Notation for GPT-2*
>
> We will add the notation for GPT-2* as the reproduced GPT-2 in the captions of Table 1-4. The training details for GPT-2* has been presented in Line 215-218.
>
> Q4: Notation for perplexity score
>
> We apologize that the expression in Line 74 may lead to misunderstanding. The minus symbol used here represents the proposed achieves a decrease on the perplexity score.
>
> Q5: The ablation study on proposed method without memory-augmentation
> We implement such ablation study via setting the hyperparameter memory-size equal to 0. Intuitively, if there is no long-term memory augmentation, the proposed method may have a performance degradation. We present the results of ablation model DEMA w/o Memory on memory-augmented in-context learning in the following Table.
>
> | Model |   In-context #Demons |  SST-2 | MR  |  Subj   |  SST-5  | MPQA | Avg |
> |   ----  |----  | ----  | ----  | ----  | ----  | ----  | ----  |
> | GPT-2*   | 4        	| $68.3_{11.6}$  | $64.7_{12.5}$  | $51.9_{4.2}$  | $31.4_{4.4}$      	| $61.5_{11.8}$      	|  55.6          |
> | MemTRM   |    4      |   $67.5_{12.4}$      	| $64.6_{11.3}$      	| $53.2_{6.0}$      	| $29.6_{4.4}$      	| $63.0_{12.1}$      	|  55.6          |
> | DEMA   | 4        	| $71.8_{14.0}$   | $65.1_{11.0}$ | $53.8_{3.7}$ | $36.0_{6.8}$ | $65.4_{12.8}$ | 58.4  |
> | DEMA  w/o Memory | 4   | $69.4_{12.4}$   | $64.3_{12.1}$ | $53.4_{7.7}$ | $29.0_{5.2}$ | $62.5_{12.3}$ | 55.7 |
> | GPT-2*   | 20  | $68.2_{11.5}$   | $63.4_{5.2}$	| $57.6_{10.2}$     	| $33.6_{6.0}$  | $70.8_{7.6}$  | 58.7      	|
> | MemTRM   | 20 | $65.1_{9.6}$    |  $65.1_{9.3}$    | $58.2_{10.6}$     	| $31.9_{6.3}$    | $72.7_{7.4}$ | 58.6 |
> | DEMA   | 20  |  $78.0_{14.1}$ | $78.6_{3.3}$  | $65.6_{8.5}$ | $36.5_{7.5}$ | $74.6_{7.3}$  | 66.7 |
> | DEMA w/o Memory |  20   |  $70.0_{12.8}$   |  $70.8_{6.2}$	| $52.9_{4.6}$	| $30.9_{6.4}$ | $72.5_{7.5}$ | 59.4 |

---

### Official Review · Reviewer_Mats · 2023-07-06

**Soundness:** 3 good
**Presentation:** 2 fair
**Contribution:** 3 good
**Rating:** 5
**Confidence:** 4

**Summary:**

The paper proposes DeMA as a learning-based adapter for existing Language Models (LMs) to increase long-term memory and performance on long inputs.

**Strengths:**

The method proposed is clear and sound and it could potentially be useful to the community since using transformers with a long context is an important open problem.

**Weaknesses:**

Experiments that prove the advantage of DeMA are limited:

- Most of the comparisons are limited to just a single memory-augmented model (MemTRM) and GPT-2 (that does not have any memory mechanism). Table 3 shows comparisons with several memory-augmented models but just on a single task and with old memory-augmented models.
- The experiments do not show comparisons with several recent memory-augmented competitors:
    - “Memformer: A Memory-Augmented Transformer for Sequence Modeling” (Wu et al.). Here a comparison on WikiText-103 might have provided additional evidence of the superiority of DeMA.
    - “Unlimiformer: Long-Range Transformers with Unlimited Length Input” (Bertsch et al.)
    - “Training Language Models with Memory Augmentation” (Zhong et al.)
    - “RWKV: Reinventing RNNs for the Transformer Era” (Peng et al.)
- I’m not convinced by the memory-augmented In-context learning experiments (Table 5). What is the purpose of this experiment? Is it really a fair “few-shot” comparison since DeMA is using additional 2000 demonstration examples loaded into cached memory? I think a comparison with other memory-augmented models will leave any doubt.

**Questions:**

- L205-206 “Then, we concatenate shuffled documents within one group and truncate them into ordered segments.” Is it not clear to me the procedure to batch the input. I also read the appendix, but it is still not clear. Could you please elaborate a little bit on this aspect?
- Why did you test your method just on the AO3 split of Chapterbreak and not on the PJ19 split?

**Limitations:**

---

> ### Author Rebuttal · Authors · 2023-08-10
>
> We thank you a lot for your detailed feedback and suggestions!
>
> Q1: Comparisons with suggested baselines
>
> Thank you so much for providing a series of competitors for comparisons. As Unlimitedformer is released within 10 days to the NeurIPS 2023 paper submission deadline, normally according to paper submission guidelines and our practical difficulties, it will not be considered as a baseline to be compared with. RWKV is a great work but it is released after the deadline of paper submission and thus impossible to be mentioned in the paper. The first pre-print version of Memformer is released in 2020 and the successor work like Memorizing Transformer used as the baseline in our paper has demonstrated more superiority as memory-augmented architectures. Additionally, we appreciate the suggestion on evaluating the method on Wikitext-103 dataset but the dataset is included as the subset of the Pile corpus used as the pre-training corpus of our work and RWKV. This is also the reason why later works pre-trained on Pile corpus did not perform evaluations on Wikitext-103.
>
> The TRIME is both a great competitor with our work and a well-recognized memory-augmented architecture. We reproduced TRIME as a memory-augmented adaptation method with the same hyperparameters of external memory as DEMA and evaluated the TRIME-long pre-trained model on the memory-augmented in-context learning tasks. The results are shown below. The TRIME still significantly lags behind the proposed method when the same number of memory-augmented demonstration examples are cached into the external memory of TRIME.
>
> | Model |   In-context #Demons |  SST-2 | MR  |  Subj   |  SST-5  | MPQA | Avg |
> |   ----  |----  | ----  | ----  | ----  | ----  | ----  | ----  |
> | GPT-2*   | 4        	| $68.3_{11.6}$  | $64.7_{12.5}$  | $51.9_{4.2}$  | $31.4_{4.4}$      	| $61.5_{11.8}$      	| 55.6      	|
> | MemTRM   |    4            |   $67.5_{12.4}$      	| $64.6_{11.3}$      	| $53.2_{6.0}$      	| $29.6_{4.4}$      	| $63.0_{12.1}$      	| 55.6      	|
> | TRIME | 4        	|    $69.5_{14.5}$  | $63.8_{9.8}$ | $51.5_{1.5}$ | $31.8_{6.7}$ | $63.6_{12.9}$ |  56.0
> | DEMA   | 4        	| $71.8_{14.0}$   | $65.1_{11.0}$ | $53.8_{3.7}$ | $36.0_{6.8}$ | $65.4_{12.8}$ | 58.4  |
> | GPT-2*   | 20       	| $68.2_{11.5}$      	| $63.4_{5.2}$       	| $57.6_{10.2}$     	| $33.6_{6.0}$      	| $70.8_{7.6}$       	| 58.7      	|
> | MemTRM   | 20 | $65.1_{9.6}$       	| $65.1_{9.3}$       	| $58.2_{10.6}$     	| $31.9_{6.3}$    | $72.7_{7.4}$ | 58.6 |
> | TRIME   | 20  |  $74.3_{13.9}$   |  $71.5_{2.5}$  |  $57.5_{11.4}$ |  $33.0_{4.6}$ | $ 69.8_{7.8}$ | 61.1  |
> | DEMA   | 20  |  $78.0_{14.1}$ | $78.6_{3.3}$  | $65.6_{8.5}$ | $36.5_{7.5}$ | $74.6_{7.3}$  | 66.7 |
>
>
>
>
> Q2: Experiments on memory-augmented in-context learning
>
> The context length of language models is usually limited to 1k-2k tokens and thus the demonstration examples used in the input prompt are limited for language models to acquire enough task representations to perform well. With the capability to utilize the demonstration examples in cached memory bank, the language models trained with our method can take advantage of even 2000 demonstration examples to gain great task representation and achieve much better performance on downstream tasks via in-context learning rather than fine-tuning. This is very crucial to the large-scale application of language models. With the augmented 2000 demonstrations, the evaluation setting is more appropriate to be described as memory-augmented many-shot in-context learning. Another significant memory-augmented model, Memorizing Transformer is also compared with our method. The results listed in Table 3 can demonstrate the superiority of our method over other memory-augmented architectures.
>
> Q3: Explanation on document batchfying process
>
> Normally, the long pre-training corpus will be truncated into fix-length segments and all segments will be randomly shuffled during training as there is no correlations between the segments in different batches. However, our proposed method requires that the i-th segment in the batch is exactly the previous context of the i-th segment in the next batch. In this way, the contextual segments in previous batches are already cached in memory bank and will benefit the language modeling towards the consecutive segments in the same document.
>
> Q4: Evaluation on only AO3 split
>
> The ChapterBreak benchmark involves two splits, the PG-19 and the AO3. The whole PG-19 corpus has been included in the pre-training corpus Pile of our model and thus cannot be deployed as the testset or validation set. The evaluation on AO3 split is appropriate to evaluate the generalization capability and zero-hot learning capability of the pre-trained language models.

---

> > ### Comment · Reviewer_Mats · 2023-08-11
> >
> > Thank you for your reply.
> >
> > > A1: Comparisons with suggested baselines
> > >
> >
> > Thank you for the additional experiments! I agree that some of the papers mentioned are too recent to be considered competitors and thus can be left out in the comparative experiments (although you should mention them in the paper). I think the paper has definitively improved from this additional experiment with a recent competitor. Is it feasible to add the comparison with TRIME also in Table 2?
> >
> > > A4: Evaluation on only AO3 split
> > >
> >
> > Thank you for the clarification. I think the paper would benefit by adding a paragraph that introduces the dataset and the decision behind choosing this split.
> >
> > Thank you again for all the other clarifications, I will update my score accordingly at the end of the rebuttal period.

---

> > > ### Author Response · Authors · 2023-08-18
> > > **Response to Reviewer Mats**
> > >
> > > Thank you so much again for kind suggestion and detailed feedback.
> > >
> > > A1: It is feasible to evaluate reproduced TRIME-long model on these language modeling benchmarks. The results are presented below:
> > >
> > > | Model |   In-context Len. |  PG22-S1  |  PG22-S2 | PG22-S3 | PG22-S4 | PG22-S5 | ArXiv |
> > > |   ----  |----  | ----  | ----  | ----  | ----  | ----  | ----  |
> > > | GPT-2*   | 1k   	|    22.78 | 24.39 | 24.12 | 24.97 | 18.07 | 11.05 |
> > > | MemTRM   |    1k     |   21.77 |  23.56 | 23.23 | 24.16 | 17.39 | 10.81 |
> > > | TRIME | 1k   	|    22.21 | 23.50 |   23.74 |   24.32   | 17.80 | 10.95 |
> > > | DEMA   |  1k     	| 21.29 | 23.01 | 22.55 |  23.35 | 16.71 | 10.05  |
> > >
> > > A2: We will add a paragraph in later version on introducing the dataset and explaining the decision behind choosing this split. Thank you so much for your kind suggestion on the paper presentation.

---

> > > > ### Comment · Reviewer_Mats · 2023-08-18
> > > >
> > > > Thank you for your response.
> > > > I've updated my score accordingly.

---

### Official Review · Reviewer_yw5j · 2023-07-07

**Soundness:** 3 good
**Presentation:** 3 good
**Contribution:** 3 good
**Rating:** 6
**Confidence:** 3

**Summary:**

This paper proposes a side network as a memory retriever and reader to help the LLMs model long context input. Specifically, the key-value pairs of the history inputs are saved in the cached memory bank. The side network takes in the current hidden states and the retrieved past key-value pairs to compute the memory-augmented representations. Their approach achieves better performance in perplexity and accuracy on multiple datasets (i.e., Project Gutenberg 2020-2022 Language Modeling Dataset, Arxiv Dataset, ChapterBreak Benchmark).

**Strengths:**

1. The side network can retrieve and fuse memory-augmented long-context, which enables the LLM to model context with ultimate length.
2. Experiments demonstrate the effectiveness of the proposed approach on long context datasets like Arxiv. Besides, this paper shows that LLM’s ability to model long context better could help both in context learning, and zero-shot performance.

**Weaknesses:**

1. The evaluation mainly focuses on perplexity score, and accuracy of suffix identification. To evaluate whether LLM models the long context well and thus benefit its’ generation ability, tasks like summarization/title generation should be included.
2. Missing comparison with Efficient Long-Text Understanding with Short-Text Models, which could take maximum 16K tokens as input.

**Questions:**

na

---

> ### Author Rebuttal · Authors · 2023-08-10
>
> We would like to thank you for your time and constructive suggestions.
>
> Q1: Evaluation on language generation tasks
>
> Thank you for the great suggestion on verifying the proposed method on language generation tasks like summarization. As the proposed DEMA model is a pre-trained language model augmented with long-term memory, the designed evaluations focus more on the emerging capabilities enabled by proposed memory-augmented pre-training, especially in-context learning or zero-shot learning. Therefore, we evaluate the proposed method on both natural language understanding and natural language generation tasks on memory-augmented in-context learning settings. The performance improvement on Squad in Table 4 can demonstrate the effectiveness of the method on language generation tasks. The document summarization or reading comprehension tasks usually require full-parameter model fine-tuning on the knowledge domain of the dataset and thus not a good fit for evaluating the capability of pre-trained language models. But we will later tune the proposed method on BookSum dataset for evaluating such domain-specific adaptation capability.
>
> Q2: Comparison with efficient long-text understanding with short-text models
>
> The baselines of LocalTRM and RoutTRM presented in Table 2 can be regarded as this type of models. And they are compared with the proposed method on long-text understanding benchmarks of ChapterBreak.

---

### Official Review · Reviewer_k2cK · 2023-07-07

**Soundness:** 3 good
**Presentation:** 3 good
**Contribution:** 3 good
**Rating:** 6
**Confidence:** 4

**Summary:**

To address the limitation of existing large language models (LLMs), the authors introduce a framework called Decoupled-Memory-Augmented LLMs (DEMA) that utilizes long-context information. DEMA utilizes a decoupled network architecture with a frozen LLM as a memory encoder and an adaptive residual side-network as a memory retriever and reader. Experimental results show that DEMA outperforms strong long-context models and achieves significant improvements in memory-augmented in-context learning over LLMs.

**Strengths:**

1.The idea of this work is novel and the motivation is strong. It tries to store and utilize long-form previous context or knowledge to address the input length limit of LLMs.

2.The main experiments seem solid as this paper verifies their method on multiple datasets and the experimental results validate the effectiveness in helping LLMs to memorize and utilize long-form information.

3.This paper is well-written and easy to follow. Their main claim is clear and easy to understand.

**Weaknesses:**

1.The paper introduces a novel lightweight residual SideNet, but there is a lack of discussion on the contribution of the SideNet module to the DEMA model in the ablation experiments.

2.In Section 3.4, the ablation experiments show that Figure 2(b) does not provide a clear visualization of the impact of different memory sizes on perplexity, especially when msz=32k (the orange line) and msz=16k (the green line) are very close to each other. This lack of distinction makes it less evident to draw a conclusive result that suggests better performance when msz=16k, as mentioned in the paper.

3.The baselines compared in the paper lack recent advancements, such as GPT-2 in 2019, GPT-3 and Bigbird in 2020, LocalTRM and RoutTRM in 2021, and MemTRM in March 2022.

4.The experiments include only main results (comparing with baselines) and the ablation studies. It would be better to conduct more detailed analyses or qualitative analysis to verify the proposed methods (e.g. whether the issues proposed in the motivation are fixed).

5.Related work section should discuss other memory augmented methods, especially the memory augment text generation methods.

6.The inputs section in Figure 1 is ambiguous and confusing. It is unclear whether the letters a, b, ..., h are meant to represent individual letters or tokens. If they represent individual letters, then combining them does not form a word. However, if they refer to tokens, it is advisable to use more precise symbols that represent tokens to avoid any ambiguity.

7.Figure2(a): sst-5 should be changed to SST-5, consistent with Table 5.

**Questions:**

In line 111, you mention ‘total # layers’. Can you explain the meaning of the "#" symbol in this context?

---

> ### Author Rebuttal · Authors · 2023-08-10
>
> We thank you a lot for your detailed feedback and suggestions!
>
> Q1: Contribution of SideNet module
>
> If the SideNet module is removed from the architecture, the model will down-grade to the architecture similar to Memorizing Transformer, which is used as the significant baseline to be compared with. Therefore, the Memorizing Transformer can be regarded as the ablation model without SideNet and the performance gap can demonstrate the contribution of proposed SideNet module.
>
> Q2: Ablation study on memory size
>
> We revised the y-axis of Figure 2(b) as the delta perplexity to show the ablation results and the new Figure is presented in the one-page rebuttal PDF document. We will revise such description on the selection of superiority of msz=16k as the hyperparameters of msz=16k and 32k present similar superiority over other hyperparameters.
>
> Q3: Other up-to-date baselines
>
> We follow the suggestions from other reviewers to add another up-to-date strong memory-augmented competitor TRIME-long[1] as the baseline and the results on memory-augmented in-context learning are shown below:
>
> | Model |   In-context #Demons |  SST-2 | MR  |  Subj   |  SST-5  | MPQA | Avg |
> |   ----  |----  | ----  | ----  | ----  | ----  | ----  | ----  |
> | GPT-2*   | 4        	| $68.3_{11.6}$  | $64.7_{12.5}$  | $51.9_{4.2}$  | $31.4_{4.4}$      	| $61.5_{11.8}$      	| 55.6      	|
> | MemTRM   |    4            |   $67.5_{12.4}$      	| $64.6_{11.3}$      	| $53.2_{6.0}$      	| $29.6_{4.4}$      	| $63.0_{12.1}$      	| 55.6      	|
> | TRIME | 4        	|	$69.5_{14.5}$  | $63.8_{9.8}$ | $51.5_{1.5}$ | $31.8_{6.7}$ | $63.6_{12.9}$ |  56.0
> | DEMA   | 4        	| $71.8_{14.0}$   | $65.1_{11.0}$ | $53.8_{3.7}$ | $36.0_{6.8}$ | $65.4_{12.8}$ | 58.4  |
> | GPT-2*   | 20       	| $68.2_{11.5}$      	| $63.4_{5.2}$       	| $57.6_{10.2}$     	| $33.6_{6.0}$      	| $70.8_{7.6}$       	| 58.7      	|
> | MemTRM   | 20 | $65.1_{9.6}$       	| $65.1_{9.3}$       	| $58.2_{10.6}$     	| $31.9_{6.3}$    | $72.7_{7.4}$ | 58.6 |
> | TRIME   | 20  |  $74.3_{13.9}$   |  $71.5_{2.5}$  |  $57.5_{11.4}$ |  $33.0_{4.6}$ | $ 69.8_{7.8}$ | 61.1  |
> | DEMA   | 20  |  $78.0_{14.1}$ | $78.6_{3.3}$  | $65.6_{8.5}$ | $36.5_{7.5}$ | $74.6_{7.3}$  | 66.7 |
>
> [1] Zhong, Zexuan, Tao Lei, and Danqi Chen. "Training Language Models with Memory Augmentation." In Proceedings of the 2022 Conference on Empirical Methods in Natural Language Processing, pp. 5657-5673. 2022.
>
>
> Q4: Additional Analysis
>
> We perform an analysis on evaluating the memory staleness issue claimed in our motivation. We take a document in the PG22 corpus as the input to Memorizing Transformer training checkpoints at 1000th steps and 1100th steps and then the average cosine similarity between the cached attention values are computed to evaluate the Memory Stableness. The cosine similarity for Memorizing Transformer is 0.81 and our proposed method is 1 because the memory encoder is frozen in our proposed model.
>
> Q6: Notation Issues
>
> The characters in Figure 1 refer to different tokens in an input sequence. We will follow your advice to revise such character symbols to actual texts to avoid the ambiguity. Additionally, the SST-5 notation in FIgure 2(a) will also be revised accordingly. At last, in line 111, he meaning of the "#" symbol is “the number of” and we will replace the symbol with textual description.

---

### Official Review · Reviewer_yoPY · 2023-07-07

**Soundness:** 3 good
**Presentation:** 3 good
**Contribution:** 3 good
**Rating:** 7
**Confidence:** 3

**Summary:**

This paper proposes a novel approach in handling long input context using memory framework. To be specific, any inputs that are longer than the maximum input size are cached in memory network (with the outcome of forward pass of LLM), and those cached memories are retrieved when "current" input is processed. The LLM remains fixed throughout training, hence being efficient; the trainable parameters are the fusion network and SideNet layers. The authors illustrate the effectiveness of the proposed approach on long-context benchmarks.

**Strengths:**

- This paper proposes to a novel approach in handling long-context problem that transformer-based language models suffer from.

- The paper evaluates the proposed idea on several benchmarks, and illustrate significant gain, especially on NLU tasks.

- The idea is sounding, and the proposed approach conducts thorough ablation studies on modules introduced.


**Weaknesses:**

- some parts of the paper need proper citations. i.e. Line 63-65: "directly adapting the entire ~~~suffers from catastrophic forgetting." needs a proper citation for the claim.

- There are baselines that could be compared against in long-context language modeling tasks. For example, in Table 2, the authors compared against MemTRM and GPT2. LocalTRM is one of the baselines that the authors can compare to in language modeling task as well, as LocalTRM has similar number of parameters and has the context length of 8K.

- It would be interested to see the inference speed of the proposed method, compared to the other baselines. It would be insightful to check how much of benefit is gained at how much expense of inference speed and computation cost.

- Much performance gains are seen over the NLU tasks, yet the gains are somewhat trivial on language modeling benchmarks (PPL).

**Questions:**

Please refer to the Weakness

**Limitations:**

Please refer to the Weakness

---

> ### Author Rebuttal · Authors · 2023-08-10
>
> We thank you a lot for your detailed feedback and suggestions!
>
> Q1: Citation for the claim for catastrophic forgetting issue
>
> We will add the paper G-MAP as a citation for this claim.
>
> Wan, Zhongwei, Yichun Yin, Wei Zhang, Jiaxin Shi, Lifeng Shang, Guangyong Chen, Xin Jiang, and Qun Liu. "G-MAP: General Memory-Augmented Pre-trained Language Model for Domain Tasks." In Proceedings of the 2022 Conference on Empirical Methods in Natural Language Processing, pp. 6585-6597. 2022.
>
> Q2: LocalTRM on Language Modeling tasks
>
> The LocalTRM adopts the relative positional bias as the positional embedding method to achieve length extrapolation. When LocalTRM is evaluated on language modeling tasks, it will down-grade to our reproduced GPT-2* which is also pre-trained language model with alibi relative positional bias.
>
>
> Q3: Inference speed and computational cost
>
> When the model is required to comprehend long sequences, the proposed method DEMA can load the out-of-boundary inputs into the cached memory as previous context. Thus, the memory usage and inference speed can be significantly improved compared with vanilla self-attention-based models.
> The detailed statistics in terms of the efficiency is presented in the following Table.
>
> | Model | In-Context Length | In-Memory Length | Inference Speed (tokens/s) | GPU-Memory Usage (MBs) |
> |--------------------------------|---------------------|--------------------|--------------------------|---------------------------|
> | GPT-2*                     	| 4k              	| N/A            	| 14666                	| 20671                 	|
> | DEMA                     	| 1k              	| 3k             	| 22638                	| 13335                 	|
> | GPT-2*                     	| 8k              	| N/A          	  | 8417                 	| 54195                 	|
> | DEMA                    	| 1k              	| 7k             	| 21343                	| 13437                 	|
>
> Q4: Perplexity Gains
>
> We believe that the gains achieved by the proposed method on language modeling benchmarks are neither trivial nor subtle. The 1.38∼1.62 perplexity decreases are remarkable performance improvements on language modeling benchmarks. Normally, over 1.0 perplexity improvement can be regarded as strong demonstrations for the effectiveness of the method, which are also recognized by other significant works like Memorizing Transformer, KNN-LM, etc.

---

### Author Rebuttal · Authors · 2023-08-10

This is the one-page rebuttal PDF document including the revisions on the Figure 1 and Figure 2 in the original paper following the suggestions from Reviewer utjk and Reviewer k2ck. We thank a lot to the detailed feedback and suggestions from all reviewers.

---

### Comment · Area_Chair_nU3s · 2023-08-19

Hi Reviewer yoPY, k2cK, yw5j, and UtJk,

Since the discussion with the authors is closing soon, could you please go over the rebuttal and provide some feedback?

Regards,

AC

---

### Decision · Program_Chairs · 2023-09-21

**Decision:**

Accept (poster)

**Comment:**

The paper proposes a method to cache the hidden state of the transformer into a memory bank and use the memory to extend the context length of large language models. Reviewers agreed on the comprehensiveness of the experiment setting and the soundness of this method.